# Home-Based Functional Electrical Stimulation of Human Permanent Denervated Muscles: A Narrative Review on Diagnostics, Managements, Results and Byproducts Revisited 2020

**DOI:** 10.3390/diagnostics10080529

**Published:** 2020-07-29

**Authors:** Helmut Kern, Ugo Carraro

**Affiliations:** 1Physiko- und Rheumatherapie, Neugebäudeplatz 1, A-3100 St. Pölten, Austria; helmut@kern-reha.at; 2Ludwig Boltzmann Institute for Rehabilitation Research, Neugebäudeplatz 1, A-3100 St. Pölten, Austria; 3Department of Biomedical Sciences, University of Padova, Via Ugo Bassi, 58/B 35131 Padova, Italy; 4Interdepartmental Research Center of Myology, University of Padova, Via Ugo Bassi, 58/B 35131 Padova, Italy; 5A&C M-C Foundation for Translational Myology, Padova, Galleria Duomo 5, 35141 Padova, Italy

**Keywords:** permanent denervated human muscle, cauda equina syndrome, home-based functional electrical stimulation, muscle co-activation, skin and muscle biopsy, color computed tomography, functional recovery

## Abstract

Spinal cord injury (SCI) produces muscle wasting that is especially severe after complete and permanent damage of lower motor neurons, as can occur in complete conus and cauda equina syndrome. Even in this worst-case scenario, mass and function of permanently denervated quadriceps muscle can be rescued by surface functional electrical stimulation using a purpose designed home-based rehabilitation strategy. Early diagnostics is a key factor in the long-term success of this management. Function of quadriceps muscle was quantitated by force measurements. Muscle gross cross-sections were evaluated by quantitative color computed tomography (CT) and muscle and skin biopsies by quantitative histology, electron microscopy, and immunohistochemistry. Two years of treatment that started earlier than 5 years from SCI produced: (a) an increase in cross-sectional area of stimulated muscles; (b) an increase in muscle fiber mean diameter; (c) improvements in ultrastructural organization; and (d) increased force output during electrical stimulation. Improvements are extended to hamstring muscles and skin. Indeed, the cushioning effect provided by recovered tissues is a major clinical benefit. It is our hope that new trials start soon, providing patients the benefits they need.

## 1. Introduction

Muscle atrophy is a muscle loss of that occurs inexorably during late aging [1,2] or earlier with prolonged malnutrition, bed rest, neural and skeletal muscle injuries, chronic cardiovascular and respiratory failures, diabetes, sepsis, and cancer. It is a dismal truth that human muscles naturally decline from the age of 30 years and there are innumerable neuromuscular disorders, which induce muscle atrophy, impairing the mobility of patients. Ill and elderly people generally spend very little time in daily physical activity. The resulting disuse atrophy limits more and more patient independence confining people to wheelchairs, beds, and to prolonged hospitalizations. Severe muscle atrophy increases risks of functional limitations, thromboembolism, and of high medical expenses [3]. Together with septic and oncologic cachexia [4], extreme sarcopenia characterizes complete, irreversible denervation of skeletal muscles, whose early stages of mild and severe atrophy evolve finally in fibro-fatty degeneration of muscle tissue [5].

Severely ill people, and even the extreme elderly, may counteract muscle deterioration by working to maintain the majority of their skeletal muscles in the best possible shape [6,7,8]. All chronic and progressive muscle impairments need permanent management. An effective and low-cost option is to educate people and patients on how to do physical activity, regardless of whether it is volitional [9] or electrical stimulation-induced exercise for those persons, which are not willing or able to move as needed [10,11]. Cardiovascular and respiratory rehabilitation for patients actually aims to reverse muscle atrophy and weakness [12,13]. Balance between costs and benefits is debated [14,15], but many practitioners recognize the value of electrical stimulation in their daily clinical practice [16,17,18,19].

Direct spinal cord stimulation promises to revolutionize the management of thoracic-level spinal cord injury (SCI) patients [20]. Unfortunately, this approach may be applied only to SCI patients with upper motor neuron lesion. While this approach deserves further validation and dissemination, there are SCI patients with complete lower motor neuron lesions (e.g., those suffering complete lesions of conus and cauda), which cannot be enrolled, i.e., those with permanent, irreversible denervation of skeletal muscles.

In the latter peculiar cases, we will show that it is possible to recover functional standing exercises by a surface home-based functional electric stimulation (hbFES) rehabilitation strategy [21]. The high intensity of the hbFES regime required to activate permanently denervated muscles is extremely painful for normal persons, and therefore is limited to patients with loss of peripheral sensation as it occurs in the complete lesion of conus and cauda equina (see below: Limitations)

Here we add that the purpose-developed electrical fields delivered to ventral aspects of the thigh muscles by anatomically shaped large surface electrodes [22] produce clinically relevant recovery of the hamstring muscles [23] and of the thigh skin [24,25,26]. A believed drawback, co-activation of non-targeted muscles, is indeed responsible for a positive clinical result.

## 2. Diagnostics

### 2.1. Patients, Assessments of Permanent Denervation, and Tissue Biopsies

#### 2.1.1. Patients

Patients suffering with an irreversible conus and cauda equina lesion (that is, up to 9.0 years of complete and permanent peripheral denervation of thigh muscles) were enrolled in the EU Program: RISE (use of electrical stimulation to restore standing in paraplegics with long-term denervated degenerated muscles (QLG5-CT-2001-02191)), following the inclusion and exclusion criteria previously reported [21]. The 25 volunteers with bilateral denervation of thigh muscles signed a consent after detailed information. They agreed to the two years of hbFES and to be submitted to tissue biopsies from both legs before and after two years of training. All applicable rules for the ethical use of human volunteers were followed during all research (approval of ethical committee, Vienna, Austria: Ethikkommission der Stadt Wien Austria: EK-02-068-0702, 5 September 2001). Clinical and functional testing and harvesting of tissue biopsies took place at the Wilhelminenspital, of the City of Vienna, Austria.

#### 2.1.2. Assessments of Permanent Denervation

Complete and permanent denervation of right and left quadriceps muscles was assessed before and after two years of hbFES by test electrical stimulation (Figure 1), needle electromyography, and both transcranial and lumbosacral magnetic stimulation, as previously described [21].

#### 2.1.3. Biopsy Harvesting

Before and after two years of hbFES, skin biopsies were collected to allow safe harvesting of muscle samples over the Vastus lateralis at the middle third of the tight [21]. Muscle biopsies were analyzed by light at the Padova University, Italy and by electron microscopy at the Chieti University, Italy. Skin biopsies fixed in formalin were embedded in paraffin at the Wilhelminenspital, Vienna, Austria. Histological sections of 5 µm thickness were collected and stained by Hematoxilin–Eosin [24,25] or by immuno-histochemistry to analyze CD1a, a biomarker of Langerhans cells [26], at the Section of Human Anatomy of the Neuroscience Department of Padova University, Italy.

### 2.2. Isometric Knee Extension Torque

We determined the force of the stimulated quadriceps muscles at each time point. Force was assessed by electrostimulation of the quadriceps muscle of patients sitting on a purpose-designed chair. The knee extension torque was assessed from a 90° knee placement by electrostimulation with a standardized stimulation program and electrodes of the hbFES [21].

### 2.3. Gross Anatomy of the Thigh Muscles and the Extent of Their Atrophy/Degeneration by Quantitative Muscle Color Computed Tomography

We determined the gross anatomy of the thigh muscles and the extent of their atrophy/degeneration using quantitative muscle color computed tomography (QMC-CT) [21,27,28,29]. QMC-CT uses the Hounsfield units for tissue characterization. Soft tissues were quantitated as subcutaneous fat, intramuscular fat, low-density muscle, normal muscle, and fibrous-dense connective tissue. To measure muscle degeneration, the pixels within defined ranges of Hounsfield units were colored (red for normal muscle, yellow for intramuscular adipose tissue, green and blue for loose and fibrous connective tissues, respectively) [29].

## 3. Managements

### hbFES of Permanent Denervated Muscles by Surface Electrical Stimulation

Two large electrodes (each 200 cm^2^) were fixed on the skin of the ventral aspect of the thigh fully covering the quadriceps muscles. The proximal electrode was near the inguinal fold, the distal near the knee with a minimal void in between. The electrical stimulation was applied to skin and muscles by a purpose developed stimulator and anatomically shaped large electrodes. They are all commercially available thanks to the generosity of Schuhfried Company, Vienna, Austria (see at http://schuhfriedmed.at/stimulette-en/stimulette-den2x-) [22]. Electrical stimulation protocol started three weeks after skin biopsy, both before and after the two years of hbFES [21].

The progressive hbFES management was personalized to the functional characteristics of the denervated muscles, i.e., mainly to SCI timespan at enrollment, never earlier than eight months from SCI. Usually, it started with 150 ms long single electrical impulses, when denervation time was longer than two years.

Patients were provided for free with stimulators and electrodes to perform hbFES for five days every week. During early training, the two large conductive-polyurethane electrodes were fixed on the thigh, by wetsponge-cloth fixed by elastic cuffs. As the skin adapted to the needed high currents, gel was placed under the polyurethane electrodes to attain minimal impedance. The electrodes were flexible enough to provide homogeneous current distribution. The training strategy consisted of four stimulation programs. At the beginning, they were biphasic stimulation impulses of long duration (120–150 ms, 60–75 ms per phase) and high intensity (up to ±80 V and up to ±250 mA). The subjects underwent clinical assessment and electrostimulated knee torque test every three months by physiatrists (Physical Medicine and Rehabilitation Physicians), who progressively modified the parameters and the training protocol according to the improvements of excitability and contractility of the thigh muscles. After the early phase of hbFES, the routine daily training consisted of twitch and tetanic stimulation patterns in sessions lasting 30 min for gluteus, thigh, and lower leg muscles on right and left legs [21].

The impulse duration was progressively shortened according to the improving excitability and contractility of the muscles. When the denervated muscles recovered the ability to respond with sustained contractions to burst of stimuli (40 ms impulse duration, 10 ms pause = 20 Hz frequency), the hbFES protocol was changed to repetitive tetanic stimulation. These tetanic contractions further improve force and mass of quadriceps muscle up to the ability to allow for standing up and stepping-in-place training [30,31,32,33]. This functional adjusted strategy (see Figure 2) is mandatory to obtain the clinically significant increase in quality and quantity of the thigh muscles [21] and the remodeling of epidermal layers of the skin [24,26].

## 4. Results

### 4.1. Results of the EU Program: RISE

The rehabilitation strategy tested by the EU RISE program led to an improvement in the thigh muscles in 20 of the 25 patients, who reached the two-year endpoint of hbFES trial [21]. They presented a statistically significant (35%) increase in cross-sectional area of the quadriceps muscle, 75% increase in diameter of quadriceps myofibers, and impressive improvement of ultrastructural organization of contractile proteins and of the Ca^2+^-handling system (T-tubules and triads) [30,31,32]. Furthermore, there was a 1187% increase in force output during electrical stimulation that allowed 25% of the endpoint patients to perform electrostimulation-assisted stand-up training [21] and Appendix A: hbFES assisted stand-up exercise.

The self-explaining and symmetric changes documented by artificially colored CT scans of both quadriceps muscle at two years of training strongly support the hbFES strategy [21]. CT scan results also showed that low compliance substantially decrease the positive results of training, but in the same subject the tridimensional mass of thigh muscles measured by 3D color CT scan increase after the patient resumed hbFES training.

Another outcome is that hbFES can be integrated into daily life without too many limitations of their normal activities. Additional benefits are the improved cosmetic appearance of lower extremities, the enhanced cushioning effect for seating, and the reduction of leg edema, as shown by changes of capillary networks observed in muscle biopsies before and after hbFES [21].

It is equally interesting to report the results at one year of hbFES [33]. Very similar increases in muscle excitability and contractility in both legs, tetanic contractions, increase of muscle mass—as shown by CT scans, that were consistently taken at the same position on both thighs, and at the same angle (Figure 3a–c)—improved appearance of limbs, and muscle cushioning. None of the subjects that performed one year of home-based daily hbFES training (20 persons) worsened their functional class, while 20% (4/20) achieved the ability to stand [33].

In short, the EU Project RISE showed that hbFES is a safe and effective management that may provide life-long physical exercise (electrical stimulation is the only option for permanent denervated muscle) recovering lost tetanic contractility of permanent denervated muscle, thus counteracting severe muscle atrophy and preventing further clinical complications.

### 4.2. Results beyond the EU Program: RISE

#### 4.2.1. Coactivation of the Hamstring Muscles

Although not appreciated by bioengineers and physiatrists, who are more interested in selective electrical stimulation to control functional movements of arms and legs, after two years of hbFES the QMC-CT cross-sectional area of the hamstrings improved, from 26.9 ± 8.4 to 30.7 ± 9.8 cm^2^, a significant 15% increase (*p* ≤ 0.05), confirming hbFES-induced muscle improvements also occurred in hamstrings (Figure 4) [23].

Comparison of left panels of Figure 4 shows the strong deterioration of degenerating muscles at one to three years of SCI. Even starting three years post-SCI, hbFES of denervated, degenerating muscles is able to recover hamstrings, although the post-hbFES analyses were performed after five years of permanent muscle denervation. This strong evidence supports co-contraction of thigh muscles as a powerful mechanism, because the induced increase of bulk of hamstrings is an effective mechanism for cushioning, and thus prevention of dermatological complications, specifically pressure sores.

#### 4.2.2. SCI-Induced Skin Atrophy Recovers after Two Years of hbFES

Using a grid, the skin thickness was measured at regular intervals in cross-sections of the skin. The mean value of the collected distances was taken as an estimate of the epidermal thickness. The area of the epidermis was also determined. Skin flattening was determined by counting the numbers of the dermal papillae per 1 mm of each image and by the interdigitation index (InterD), an index of the complexity of the interdigitation within the epidermal–dermal junction. The ID is 1.00 when the skin is flat, while it increases with the complexity of the dermal papillae reaching a value of 1.30 in young people [24]. Linear regression analyses of skin characteristics show that the epidermis progressively decreases in thickness and in dermal papillae complexity [24]. On the other hand, two additional years of SCI and hbFES recovers the properties of the epidermis to those presented in the skin biopsied earlier than one year of SCI [26].

#### 4.2.3. Rise2-Italy Project: Functional Echomyography

Mono-lateral denervated leg muscles were analyzed by Functional Echomyography to monitor thickness, contraction–relaxation properties, and perfusion characteristics during and after electrical stimulation [34]. Morphology and ultrasonographic structure changed during several months of hbFES from complete denervated muscle to the one that is “normal” in old patients. Contraction–relaxation kinetics, during electrical stimulation, showed a significantly longer relaxation phase in the denervated muscle. Echo doppler analyses at rest demonstrated a low-resistance arterial flow that pulsed during electrostimulation. The hyperemia lasted longer than stimulation. Anyhow, the increased blood flow in response to electrical stimulation (either creating a direct blood volume effect, or through “feeding” muscle cells for growth) seems to be a rational mechanism related to the positive trophic effect induced by hbFES management. Furthermore, the higher than normal energy needed to activate the muscles demonstrate that the muscles remained denervated during the one-year of training. In conclusion, Functional Echomyography is a useful tool in the follow-up of hbFES for denervated muscles, as it is well known for the ultrasound imaging approaches in aging myopathy [35] and in tendinopathies [36].

### 4.3. Advantages of or Limitations to the Outcomes

#### 4.3.1. Advantages

Patients suffering with thoracic, lumbar, and sacral SCI use wheelchairs to gain some independence. A wheelchair implies that a person squeezes hamstrings and gluteal muscles several hours daily, sadly contributing to more severe atrophy of the muscles, of leg edema, and osteoporosis, with risks of decubitus, thrombophlebitis, and fractures [37]. However, of particular importance in SCI is whether the connection between muscle and nerve is preserved or if muscles are irreversibly denervated, as it may occur in complete lesion of conus and cauda equina. In the first case (see for review [38]), soon after SCI, there is a rapid loss of muscle mass. At six weeks post-SCI, muscle cross-sectional areas were decreased by between 18% to 46%. Prospective study up to 24 weeks post-SCI demonstrated declines of gastrocnemius and soleus muscles of 24% and 12%. From six weeks to 24 weeks the decrease in quadriceps, hamstrings, and adductor muscle cross section areas (CSA) were around 15%. Advancing age and duration of injury have been associated with less percentage of muscle tissue, but a study comparing two-year versus twenty-years SCI demonstrated a substantial stability of leg muscle atrophy in long-term paraplegics with complete upper motor neuron lesion. Eventually, further problems emerge for these patients due to excessive contractions in the spastic muscles [39].

In the latter cases, the irreversibly denervated muscle becomes unexcitable with commercial electrical stimulators used for innervated muscles (both in normal adults or thoracic-level SCI patients) because they undergo ultrastructural disorganization within a few months, and then to severe atrophy (up to severe sarcopenia) with muscle fibers displaying nuclear clumping, sparse regeneration [30,31], and finally fibro-fatty full degeneration within 3–6 years from SCI [30,31,32,33]. Luckily, the process of muscle fiber degeneration is slow enough to allow the person in need to start hbFES Vienna Strategy in time to reverse the process and re-build almost normal-like muscle fibers with the characteristics of the fast type human muscle fibers [21,30,31,32]. Indeed, here we reviewed data demonstrating that electrical stimulation by very large electrodes covering the full quadriceps muscle, to activate the entire population of denervated muscle fibers of the human thigh, is able to recover both ventral (quadriceps muscles) and dorsal (hamstrings muscle group) muscles of the thigh. Furthermore, QMC-CT of thigh muscles treated for two years with hbFES strongly support the efficacy of the Vienna Strategy for hbFES of denervated, degenerating muscles [23]. A noteworthy point is that the color CT scans taken at enrollment (before any hbFES) seem to suggest that the hamstrings are more atrophic and degenerating than the quadriceps muscles [21,30]. It is thus not surprising that the effects of hbFES are more evident in the quadriceps than in the co-activated hamstrings. Nonetheless, the improvements noted in the hamstrings muscle group are a clinically relevant effect of hbFES because the recovered tissues of the dorsal thigh muscle contribute an important cushioning effect for sitting position. Furthermore, because we harvested muscle tissue through a skin biopsy, we extend our analyses to skin. Excitingly, we found that the home-based functional electrical stimulation-sustained improvements in muscle structures are extended to skin, whose thickness and flattening worsen in accordance with the extent of years of SCI [24], but are substantially recovered after two years of hbFES [26]. The increase of tissue trophism will be clinically relevant if hbFES is extended to gluteal muscles for their training workout, as the Vienna Strategy strongly recommends. On the other hand, this prolongs the hours per day that the patients dedicate to muscle training.

#### 4.3.2. Main limitation

The only limitation of hbFES for denervated, degenerating muscles is that it can only be applied in case of completely damaged conus and cauda equina, or in case of complete transverse SCI, because degenerating muscles respond only to very high intensity electrical stimulation, unacceptably painful for SCI people with residual innervation or reinnervation.

## 5. Byproducts

### 5.1. Neuromuscular Electrical Stimulation Appropriate for Elderly People

Standing on the results of the EU RISE Program, the Colleagues of Vienna, Austria designed and realized stimulators for home-based neuromuscular electrical stimulation appropriate for elderly people. As described in Kern et al. [40], elderly persons performed neuromuscular electrical stimulation training at home. Results demonstrated that when training for nine weeks (two times a week 3 × 10 min for each session), neuromuscular electrical stimulation for elderly persons is safe and effective. All subjects achieved neuromuscular electrical stimulation-induced full knee extension. The outcomes included a significant increase in muscle strength and an increase in the number and size of fast myofibers, that are the first to respond to electrical stimulation and whose content and size are correlated to muscle strength. In muscle biopsies, Pax7- and NCAM-positive muscle satellite cells were also increased in the absence of muscle damage and cellular inflammation [41,42,43,44,45,46,47,48,49,50,51]. Finally, there are many applications of in-home or in-hospital functional electrical stimulation managements of disuse muscle atrophy related to organ diseases, from chronic cardiovascular failures to functional electrostimulation cycling in SCI [52,53,54,55,56].

### 5.2. Machine Learning Predictive Systems

A further byproduct is a predictive system based on radiodensitometric distributions from mid-thigh CT images. Muscle deterioration in elderly individuals is commonly characterized by loss of muscle strength and replacement of lean tissue mass with inter- and intra-muscular adipose tissue. These phenomena are independent mortality risks in aging individuals. The incidence of muscle deterioration in aging, commonly referred to as sarcopenia, significantly affects the quality of life and physical activity of aging individuals [57]. Machine learning (ML) algorithms, are becoming increasingly used in healthcare data applications. The increased availability of healthcare data and the continued development of big data analytics methods has driven the success of ML modelling in many quantitative fields, such as medical image processing or predictive system development, as well as other specialties such as neurology, cardiology, and oncology [58,59,60,61,62]. Mid-thigh computed tomography (CT) images from the Age, Gene/Environment Susceptibility (AGES) dataset have been used to quantitatively characterize subject-specific changes in soft tissue using a novel method known as non-linear trimodal regression analysis (NTRA) [63]. The NTRA method works by generating soft tissue regression profiles described by 11 unique NTRA model parameters. The utility of these parameters in quantifying differences in fat, lean muscle, and loose connective tissue was first explored in comparing young, aging, and pathological subjects. Results from this work illustrated the sensitivity of NTRA parameters to changes in soft tissue and suggested the employment of this method in the context of a larger CT image database. The Age Gene/Environment Susceptibility Study (AGES-Reykjavík) is an Icelandic dataset designed to examine risk factors and gene/environment interactions in relation to disease and disability in aging people. This dataset was assembled using 3152 volunteers from 66–92 years of age analyzed at two time points separated by five years. The AGES-Reykjavík dataset thereby presents a unique opportunity to employ big data analytics methods such as ML modelling [64]. As ML algorithms have illustrated strong predictive value in the regression of body mass index (BMI) and isometric leg strength (ISO), the study sought to demonstrate their prediction using NTRA parameters obtained from CT mid-femur cross-sections in the AGES-Reykjavík dataset. Results from this work further consolidate the predictive power of NTRA using BMI and ISO as test parameters. The method is useful in prediction studies of mobility and cardiocirculatory diseases [27,65]. Further investigations of the correlations between these parameters and the consequent risks could further expand the field of translational myology to sarcopenic muscle degeneration and its downstream effects on the health of the elderly.

## 6. Conclusions

We demonstrated the long-term clinical value of co-activating thigh muscles through hbFES strategy using high currents and large surface electrodes. This Vienna Strategy is able to reverse, at clinically relevant levels, the adverse effects of SCI, even in the worst-case scenario of complete lesion of conus and cauda equina. Continued regularly, hbFES for denervated, degenerating muscles helps to maintain healthier leg muscles and skin, reducing the risks of life-threatening SCI complications.

We hope that international physiatrists will collaborate in new independent multi-center trials for denervated, degenerating muscles recovery by hbFES. Involvement of physiatrists will be helpful in other centers worldwide, including where there is already interest, such as in Russia [66,67,68,69,70], to offer to SCI patients with complete lesions of conus and cauda equina the opportunity they deserve.

## Figures and Tables

**Figure 1 diagnostics-10-00529-f001:**
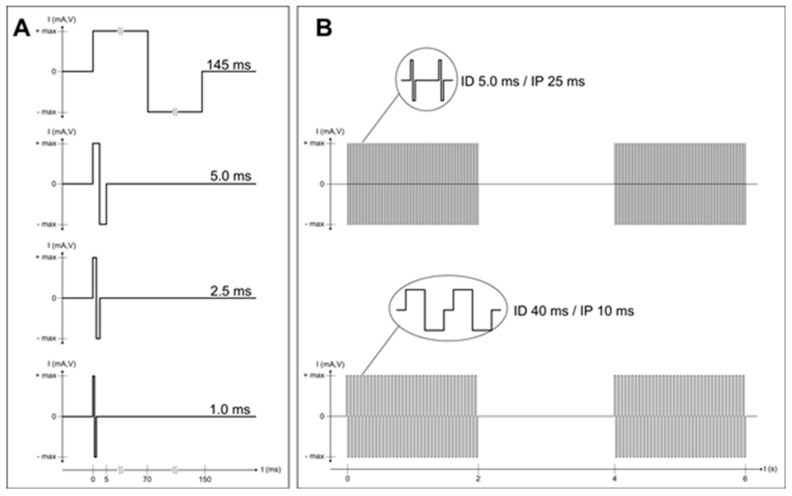
Complete and permanent denervation of quadriceps muscles has been confirmed by test electrical stimulation before and after two years of hbFES. Panel (**A**) shows the single twitch characteristics of the electrical stimulation test at the enrolment of SCI patients in the EU RISE Project. Normal innervated muscles respond at 1.0 ms twitch stimulation. Denervated muscles tested days or weeks after SCI may respond to 2.5 ms or 5.0 ms twitch stimulation. Patients with complete muscle denervation tested 12 or more months after SCI may only respond very weakly to 145 ms long impulses. Three to five years after SCI, muscles seem to be unexcitable even using electrical stimuli longer than 150 ms, but after several (4–12) weeks of hbFES Vienna training, permanent denervated muscles recover contractility. During the first months of hbFES training the denervated myofibers respond to shorter and shorter test stimuli [21], but even the best responder SCI patients never return able to respond to test stimuli shorter than 40 ms (panel (**B**)), not only with macroscopic contractions, but also with minimal contractions checked by ultrasound analysis, i.e., by functional echomyography.

**Figure 2 diagnostics-10-00529-f002:**
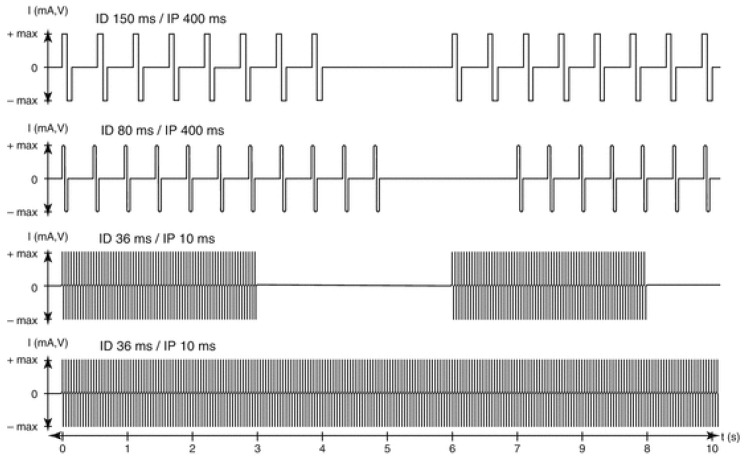
Parameters for progressive hbFES strategy of long-term permanently denervated human muscles. The progressive training starts, usually more than nine months after SCI, with bursts of stimulation duration (SD) of 4 s and a stimulation pause (SP) of 2 s. Notice that the burst contains impulses with an impulse duration (ID) of 150 ms and an impulse pause (IP) of 400 ms to activate severely atrophic muscle fibers, able to respond only with twitch contractions. This first period of conditioning last usually 3–5 months, but could be shortened, if the hbFES starts earlier than six months from SCI. When at successive follow-up the muscle presents higher excitability and stronger twitches, the ID can be reduced to 80 ms and SD increased to 5 s for another four months. When in the following follow-up test stimulation shows that the muscle respond to ID of about 40 ms, the next training phase implement tetanic burst stimulation of 3 s SD and 3 s SP with impulses of 40 ms ID and 10 ms IP (20 Hz frequency). The induced tetanic contractions are potent trophic factors that increase muscle mass, density and diameter of muscle fiber and then force, inducing in sitting position leg extensions without and then with additional weights on the ankles of the patients (progressive resistance training). When a good muscle condition is achieved, which depends not only to the training intensity but also to the time span of denervation, standing, stepping-in-place, and walking exercises can be performed using continuous stimulation (controlled by an external switch) with 40 ms ID and 10 ms IP.

**Figure 3 diagnostics-10-00529-f003:**
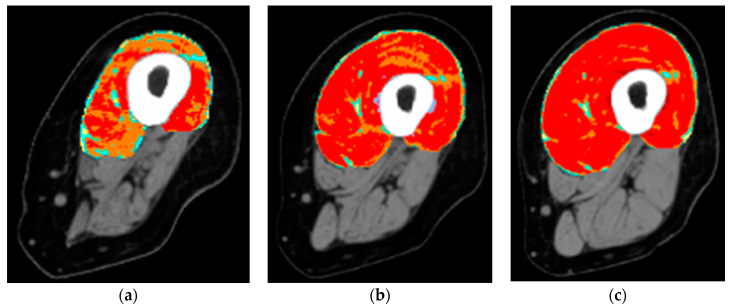
Colored cross-sectional area of the quadriceps muscle before and after one or two years of hbFES. In a 49-year-old female patient, 2.5 years after denervation, at enrollment (**a**), after one year (**b**), and after two years (**c**) hbFES. The increase of muscle bulk, as shown by color CT scan analyses, is clearly present after one year of hbFES and continues during the second year of training. The muscle recovery is present also in the hamstrings, in particular after two years of hbFES. Color code for the quadriceps muscle: red = normal muscle, orange = atrophic muscle, blue = connective tissue, yellow = intramuscular fat.

**Figure 4 diagnostics-10-00529-f004:**
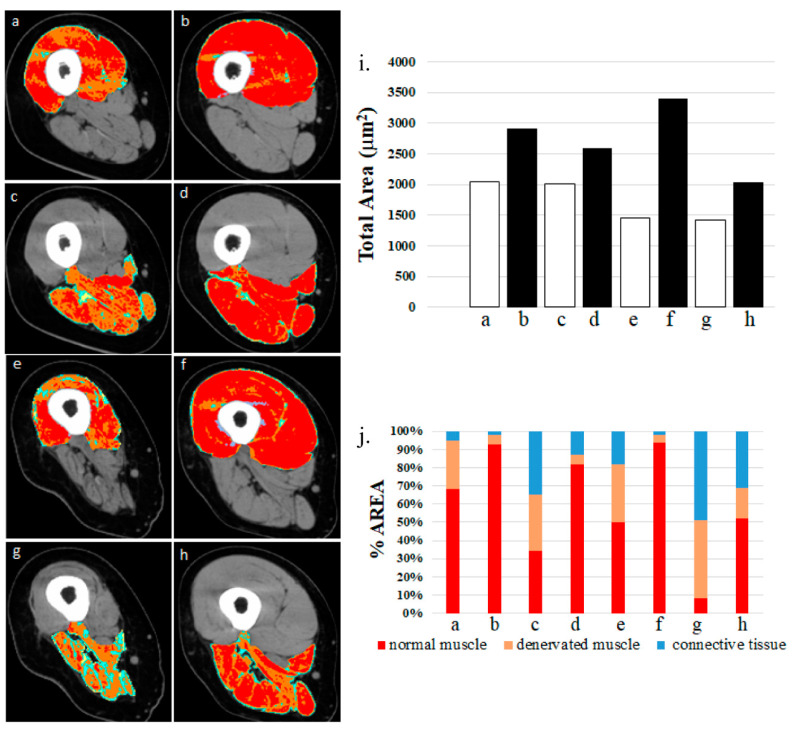
Denervated quadriceps and hamstring muscles respond to hbFES years after SCI. Patient A: (**a**) quadriceps muscle at one year post-SCI, no hbFES; (**b**) muscle in panel “a” after two years of hbFES. Patient B: (**c**) hamstring muscle at one year post-SCI, no hbFES; (**d**) muscle in panel “c” after two years of hbFES; (**e**) Patient C: quadriceps muscle at three years post-SCI, no hbFES; (**f**) muscle in panel “e” after two years of hbFES. Patient D: (**g**) hamstring muscle at three years post-SCI, no hbFES; (**h**) muscle in panel “g” after two years of hbFES (**i**). Total area of muscles a–h (**j**). Quantitative computed densitometric analyses of tomography, muscles a–h. (Reproduced with permission from [23]).

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
