# Peer review of "Home-Based Functional Electrical Stimulation of Human Permanent Denervated Muscles: A Narrative Review on Diagnostics, Managements, Results and Byproducts Revisited 2020"

_diagnostics, 2020, doi:10.3390/diagnostics10080529_

Round 1

Reviewer 1 Report

Kern and Carraro's review is well-written and clear.

This review describes, for patients with spinal cord injuries, the benefits of electrical stimulation of the lower limbs to limit atrophy and recover muscle mass. This maintenance of muscle mass has many clinical benefits for patients such as improved blood circulation and reduced susceptibility to pressure ulcers.

However, an error has crept in (line 323). The 24 training sessions are not carried out over 1 week but over 9 weeks...

Author Response

Following the only request of changes of the Referee # 1, we added at the line 323:

for 9 weeks (two times a week),

Reviewer 2 Report

hbFES of human permanent denervated muscles:  diagnostics, managements, results and byproducts revisited 2020.         Kern & Carraro.   July 2020.

Listed as a review article, this paper primarly presents the authors’ previous results on thigh muscles and skin of Home-based Functional Electrical Stimulation (hbFES) for 25 patients with denervation atrophy of quadriceps and hamstrings consequent on spinal cord injury (SCI) from complete conus or cauda equina syndromes.  Unfortunately at present, much of the paper is written as though it were a research article presenting those studies and results, rather than as a review of the field.

Major Points

  1. The authors refer primarily to several previous papers from their own centres, and it seems that the work underlying this review was predominantly performed and written-up in those papers, particularly in ref.21 (Kern H., Carraro U. et al 2010), illustrating the quadriceps findings, and in subsequent papers illustrating the hamstring and skin findings. There does not appear to be much mention or discussion of work outside those centres, and no new material in the present paper.

  2. This review of hbFES will be of interest to health care professionals who have patients with chronic conditions associated with denervation/muscle atrophy, quite apart from patients post-SCI; but would be of most value if based on assessment of available published results from all different centres and different patient conditions and circumstances, where applicable. For example, at the very end of the article, in ‘Conclusions’ the authors do give the references of five publications from Russia (refs. 68-72), but provide no data, results or discussion from these. These may be articles for which evaluation could contribute to a review, especially as they seem currently inaccessible to most readers – written in Russian, and in a journal (Bulletin of Rehabilitation Medicine) which is not found on an internet search or through ‘PubMed’; and the authors should attend to this.

  3. The authors do in section 5.1 and 5.2 mention wider application of hbFES, and of analysis of mid-femur CT scan cross-sections for sarcopenia assessment with ageing, but at present these two paragraphs seem rather as add-ons, and disconnected from the thread of the rest of the paper, rather than being a component part of a considered critical review. There is, for example, also other literature discussing hbFES in muscle wasting associated with COPD or with chronic peritoneal dialysis.

  4. Therefore, overall, the paper needs to be re-written in the style of a critical summary review of the field, rather than as a repetition of the authors’ description of their previous work. The authors may wish to adjust the title accordingly to reflect this : eg. ‘Home-based Functional Electrical Stimulation (hbFES) of human permanently denervated muscles, and of sarcopenic or other wasted muscle – a Review’

More minor Points
There are also some more detailed specific points in the present manuscript which would require attention.
These are :

  1. Lines 53-56 : Since this paper is a review article, and therefore may be read by non-specialists, the authors need to confirm here that the study group in their series (patients with irreversible denervation) are patients with LMN (lower motor neuron) lesions, whereas the thoracic-level SCI group will be ones with UMN (upper motor neuron) lesions.
     They also need to give the total number of patients contributing to the study (? 25) and to say what proportion of their patient group have unilateral or bilateral involvement, in comparison with the proportions in patients with paraplegia from SCI.

  2. Line 61-62 : The authors need to add a comment here that the high intensity of the hbFES regime required is extremely painful, and therefore is limited to patients with loss of peripheral sensation (see lines 315-6).

  3. Line 61 : ‘believed’ rather than ‘belived’

  4. Line 68-70 : The RISE program is titled as being for paraplegics with denervated muscles. If any of the patients were affected by denervation unilaterally, rather than bilaterally, this needs to be stated and quantified here (see line 259, which refers to ‘contralateral normally innervated muscle’).

  5. Line 92 : ‘Figure 1. and permanent denervation….’ Is there a word missing here ?

  6. Line 97 : ‘weakly’ rather than ‘weekly’.

  7. Line 156 : Should this be : ‘…wet sponge-cloth…’  rather than ‘…wet sponge clothes…’  ?

  8. Line 162 : The term ‘Physiatrists’ may not yet be universally recognised. I would suggest adding :
    ‘…physiatrists (Physical Medicine and Rehabilitation Physicians)…’

  9. Line 200 : ‘…as shown by CT scan analysis…’ What measures were taken to guarantee that the CT scans 1 and 2 years apart are consistently taken at the same position on the thigh, and at the same angle in relation to it ?

  10. Line 201-203 : Were there any subject controls – eg. patients or opposite affected limb receiving no treatment or perhaps passive movements of the limb without electrical stimulation ?

  11. Lines 225-231 : Please confirm here that frames a,b,c,d are all one patient, whereas frames, e,f,g,h are all a 2nd different patient. This needs making clear in the Legend :  eg.  ‘Patient A……; Patient B…… ‘

  12. Line 232-3 : The CT scans as the Left panels in Fig. 4 only provides evidence in these 2 patients for degeneration, and only then if a normal non-degenerated muscle appearance is also shown. The authors may like to show a normal section of thigh as baseline comparison.

  13. Line 250 : Were those changes observed over time in skin compared with any control skin biopsy (from, say, the opposite limb) taken at the same time ?.

  14. Line 259 : As in Point 8 above, please specify in section 2.1.1 (Patient description) how many patients were affected unilaterally by the SCI, and how many bilaterally; and if they were bilateral whether they had complete or partial denervation bilaterally. In how many could the ‘unaffected leg’ be taken as a true control for both skin and muscle changes rather than having partial denervation in that leg ?.

  15. Line 261-2 : The authors need to say here whether a pulsed arterial flow implies an improvement in blood supply, and if so, whether they can tell if increase in muscle bulk is a direct response of muscle cells to electrical stimulation, or whether it might be secondary to increased blood flow in response to electrical stimulation (either creating a direct blood volume effect, or through 'feeding' muscle cells for growth) ?

  16. Line 288 : ‘…undergo…’ rather than ‘…undergoes…’

  17. Line 291 : ‘…luckily…’ rather than ‘…luckily…’

  18. Line 315-6 : See Point 6 above.

  19. Line 327 : ‘…strength…’ rather than ‘…streght…’

  20. Lines 333-363 (section 5.2) This paragraph seems at present to be an add-on largely unrelated to the rest of the paper. If the intention is for this paper to be an overview of CT investigation and electrical-stimulation management of muscle atrophy from different causes (including as sarcopenia in the elderly) then the paper needs to be rewritten as such. If the paper is specifically about hbFES, then this paragraph may be better shortened to being a statement of the role of CTscan  and hbFES compared with other approaches in elderly sarcopenia.

  21. Lines 370-2.  See Point 2 above.  There has been no previous mention in this paper of the involvement or contribution of Russian centres to in the field of hbFES; and if there is significant contributory data in the referenced publications (refs. 68-72), this should be discussed earlier in the paper.  I would suggest that this sentence be reduced merely to mention that multi-centre trials and involvement of physiatrists will be helpful in other centres worldwide, including where there is already interest, such as in Russia (refs.68-72, and perhaps the supplementary material video ?).

  22. Lines 2-4 : Title of paper. ‘…permanently…’ would be better than ‘…permanent…’
    … but please also see Point 4 above.

Author Response

REPLY TO THE Major criticisms of the referee 2.

The Referee # 2 unfortunately did not recognized (or understood, due to our poor English) that our NARRATIVE REVIEW reports the ONLY RESULTS existing in the world letterature related to the diagnostics and management by h-bFES of PERMANENT LONG-TERM DENERVATED LEG MUSCLES secondary to a complete SCI lesion of the Conus Cauda. This is a problem that we had to overcome in all our long series of related publications.

Referee #2 Comments 1 to 4:

  1. The authors refer primarily to several previous papers from their own centres, and it seems that the work underlying this review was predominantly performed and written-up in those papers, particularly in ref.21 (Kern H., Carraro U. et al 2010), illustrating the quadriceps findings, and in subsequent papers illustrating the hamstring and skin findings. There does not appear to be much mention or discussion of work outside those centres, and no new material in the present paper.
  2. This review of hbFES will be of interest to health care professionals who have patients with chronic conditions associated with denervation/muscle atrophy, quite apart from patients post-SCI; but would be of most value if based on assessment of available published results from all different centres and different patient conditions and circumstances, where applicable. For example, at the very end of the article, in ‘Conclusions’ the authors do give the references of five publications from Russia (refs. 68-72), but provide no data, results or discussion from these. These may be articles for which evaluation could contribute to a review, especially as they seem currently inaccessible to most readers – written in Russian, and in a journal (Bulletin of Rehabilitation Medicine) which is not found on an internet search or through ‘PubMed’; and the authors should attend to this.

  3. The authors do in section 5.1 and 5.2 mention wider application of hbFES, and of analysis of mid-femur CT scan cross-sections for sarcopenia assessment with ageing, but at present these two paragraphs seem rather as add-ons, and disconnected from the thread of the rest of the paper, rather than being a component part of a considered critical review. There is, for example, also other literature discussing hbFES in muscle wasting associated with COPD or with chronic peritoneal dialysis.

  4. Therefore, overall, the paper needs to be re-written in the style of a critical summary review of the field, rather than as a repetition of the authors’ description of their previous work. The authors may wish to adjust the title accordingly to reflect this : eg. ‘Home-based Functional Electrical Stimulation (hbFES) of human permanently denervated muscles, and of sarcopenic or other wasted muscle – a Review’

We understand the previous four major criticisms and suggestions, but it is IMPOSSIBLE to comply (see above). Thus, we will ONLY accept all following suggestions, for which we thank a very patient and expert Referee #2.

Referee #2 Comment 5:

5. Lines 53-56 : Since this paper is a review article, and therefore may be read by non-specialists, the authors need to confirm here that the study group in their series (patients with irreversible denervation) are patients with LMN (lower motor neuron) lesions, whereas the thoracic-level SCI group will be ones with UMN (upper motor neuron) lesions. They also need to give the total number of patients contributing to the study (? 25) and to say what proportion of their patient group have unilateral or bilateral involvement, in comparison with the proportions in patients with paraplegia from SCI.

We added as suggested at the line (now 53-58):

Unfortunately, this approach may be applied only to SCI patients with upper motor neuron lesion. While this approach deserve further validation and dissemination, there are SCI patients with complete lower motor neuron lesions (e.g., those suffering complete lesion of Conus and Cauda Equina), which cannot be enrolled, i.e., those with permanent, irreversible denervation of skeletal muscles.

Referee #2 Comment 6:

Line 61-62 : The authors need to add a comment here that the high intensity of the hbFES regime required is extremely painful, and therefore is limited to patients with loss of peripheral sensation (see lines 315-6).

As suggested we added at the Lines (now) 64-65:

The high intensity of the hbFES regime required to activate permanently denervated muscles is extremely painful for normal persons, and therefore is limited to patients with loss of peripheral sensation as it occurs in the complete lesion of Conus and Cauda Equina (see below: Limitations).

Referee #2 Comment 7:

Line 61 : ‘believed’ rather than ‘belived’

DONE. Thanks.

Referee #2 Comment 8:

Line 68-70 : The RISE program is titled as being for paraplegics with denervated muscles. If any of the patients were affected by denervation unilaterally, rather than bilaterally, this needs to be stated and quantified here (see line 259, which refers to ‘contralateral normally innervated muscle’).

At the now line 75 we added:

The 25 volunteers with bilateral denervation of thigh muscles

The patients referred at the line 259 were NOT RISE patients, but patients enrolled in a following Italian Research Project.

Referee #2 Comment 9:

Line 92 : ‘Figure 1. and permanent denervation….’ Is there a word missing here ?

YES, thank you. We added: at the now line 98:

Figure 1. Complete and permanent

Referee #2 Comment 10:

Line 97 : ‘weakly’ rather than ‘weekly’.

YES, thank you.

We changed at the now line 103 to: weakly

Referee #2 Comment 11:

Line 156 : Should this be : ‘…wet sponge-cloth…’  rather than ‘…wet sponge clothes…’  ?

YES, thank you, we changed at the now line 162:

using wet sponge-cloth fixed

Referee #2 Comment 12:

YES, thank you, we changed at now line 168 to:

Physical Medicine and Rehabilitation Physicians

Referee #2 Comment 13:

Line 200 : ‘…as shown by CT scan analysis…’ What measures were taken to guarantee that the CT scans 1 and 2 years apart are consistently taken at the same position on the thigh, and at the same angle in relation to it ?

YES, thank you, we added at now line 207:

taken at the same position on the thigh, and at the same angle

Referee #2 Comment 14:

Line 201-203 : Were there any subject controls – eg. patients or opposite affected limb receiving no treatment or perhaps passive movements of the limb without electrical stimulation ?

Though not included as control, several patients that refused h-bFES were followed-up in Vienna.

No one improved in the following years ... 

We thanks the referee, may be a study on those patients will be interesting NOW, but those patients are rare and usually are difficult to contact, if NOT under h-bFES.

Referee #2 Comment 15:

Lines 225-231 : Please confirm here that frames a,b,c,d are all one patient, whereas frames, e,f,g,h are all a 2nd different patient. This needs making clear in the Legend :  eg.  ‘Patient A……; Patient B…… ‘

YES, thank you: In the now lines 232-236 we added:

Patient A: ....; Patient B: ....; Patient C: .... Patient D: ....;

Referee #2 Comment 16:

Line 232-3 : The CT scans as the Left panels in Fig. 4 only provides evidence in these 2 patients for degeneration, and only then if a normal non-degenerated muscle appearance is also shown. The authors may like to show a normal section of thigh as baseline comparison.

In this case we do not agree, because the variability of the thigh section is very large ALSO in normal persons and thus meaningless as comparison for these cases.

Referee #2 Comment 17:

Line 250 : Were those changes observed over time in skin compared with any control skin biopsy (from, say, the opposite limb) taken at the same time ?

Again we do not agree, because the variability of the skin thickness is very large ALSO in normal persons and thus meaningless as comparison for these cases. Furthermore it is NOT easy to collect skin biopsies from normal persons for ethical reasons ...

Referee #2 Comment 18:

Line 259 : As in Point 8 above, please specify in section 2.1.1 (Patient description) how many patients were affected unilaterally by the SCI, and how many bilaterally; and if they were bilateral whether they had complete or partial denervation bilaterally. In how many could the ‘unaffected leg’ be taken as a true control for both skin and muscle changes rather than having partial denervation in that leg ?

We do not comply with these suggestions, because the referee is mixing requests for the EU RISE Project patients with those of the "Results beyond the EU Program: RISE", that is one of the byproducts of the EU RISE Project.

Referee #2 Comment 19:

Line 261-2 : The authors need to say here whether a pulsed arterial flow implies an improvement in blood supply, and if so, whether they can tell if increase in muscle bulk is a direct response of muscle cells to electrical stimulation, or whether it might be secondary to increased blood flow in response to electrical stimulation (either creating a direct blood volume effect, or through 'feeding' muscle cells for growth)?

YES, thank you. We added at the now line 270:

Anyhow, the increased blood flow in response to electrical stimulation (either creating a direct blood volume effect, or through 'feeding' muscle cells for growth) seems to be a rational mechanism related to the positive trophic effect induced by hbFES management.

Referee #2 Comment 20:

Line 288 : ‘…undergo…’ rather than ‘…undergoes…’

YES, Thank you. At the now Lines 298, we changed as suggested.

Referee #2 Comment 21:

Line 291 : ‘…luckily…’ rather than ‘…luckily…’

YES, Thank you. At the now Line 301, we changed as suggested.

Referee #2 Comment 22:

Line 315-6 : See Point 6 above.

YES, thank you. As suggested at the Point 6, we stressed at now the lines 64-65 that: "The high intensity of the hbFES regime required to activate permanently denervated muscles is extremely painful for normal persons, and therefore is limited to patients with loss of peripheral sensation as it occurs in the complete lesion of Conus and Cauda Equina."

Referee #2 Comment 23:

Line 327 : ‘…strength…’ rather than ‘…streght…’

YES, Thank you. At the now Lines 298, we changed as suggested.

Referee #2 Comment 24:

Lines 333-363 (section 5.2) This paragraph seems at present to be an add-on largely unrelated to the rest of the paper. If the intention is for this paper to be an overview of CT investigation and electrical-stimulation management of muscle atrophy from different causes (including as sarcopenia in the elderly) then the paper needs to be rewritten as such. If the paper is specifically about hbFES, then this paragraph may be better shortened to being a statement of the role of CTscan  and hbFES compared with other approaches in elderly sarcopenia.

We do not agree with the Referee. Because we are listing one of the key "Byproducts" of the EU RISE Project, we defend our opinion to do not modify the typescript at the now lines: 343-373.

Referee #2 Comment 25:

Lines 370-2.  See Point 2 above.  There has been no previous mention in this paper of the involvement or contribution of Russian centres to in the field of hbFES; and if there is significant contributory data in the referenced publications (refs. 68-72), this should be discussed earlier in the paper.  I would suggest that this sentence be reduced merely to mention that multi-centre trials and involvement of physiatrists will be helpful in other centres worldwide, including where there is already interest, such as in Russia (refs.68-72, and perhaps the supplementary material video ?).

YES, thank you. We modify accordingly the sentence, now at the lines 380-384:

We hope that international physiatrists will collaborate in new independent multi-centre trials for denervated, degenerating muscles recovery by h-bFES. Involvement of physiatrists will be helpful in other centres worldwide, including where there is already interest, such as in Russia [68–72],  to offer to SCI persons suffering with complete conus and cauda equina syndrome the opportunity they deserve.

Referee #2 Comment 26:

Lines 2-4 : Title of paper. ‘…permanently…’ would be better than ‘…permanent…’
… but please also see Point 4 above.

YES thanks you. We modify the TITLE as here suggested, but we like to defend our overall original TITLE, stressing the fact that it is a NARRATIVE REVIEW.

New Title:

Home-based Functional Electrical Stimulation of human permanent denervated muscles: A narrative review on diagnostics, managements, results and byproducts revisited 2020

Round 2

Reviewer 2 Report

Home-based Functional Electrical Stimulation of
3 human permanent denervated muscles: .........
Kern H. & Carraro U.

25.07.20   Comments on revision (vs2.)

In submitting this revised version of their review article, the authors have defended their overall approach by affirming that theirs are the only centres who have published on this technique.  Accepting this, the paper does not then require the suggested ‘major revision’; while the style of writing as a re-run of the authors experimental results, also becomes more acceptable.  

However, in clarifying in their comments to referee 2, that RISE and RISE2-Italy, involve different subjects, the authors need not only to make this clear also in the paper, but to define the several different patient groups on which are based the additional findings of changes in blood flow, changes in skin, in  hamstring muscle properties, and in sarcopenic muscle in the elderly. 

Please see the details in the numbered points below, covering these issues.

Also, unfortunately, many of the ‘edits’ made to version 1 in response to the reviews, have themselves introduced further typographical errors (mostly through failure to separate words) which now need re-correction, as listed after the five numbered points.  

Point 1.
Lines 241-255 .  Section 4.2.3

The authors have clarified in their comments to referee 2 that the unilateral denervated patients were in the RISE2-Italy Project (ref. 35), who were separate from the 25 bilateral patients in the original RISE Project.  The authors must indicate this in this paper, but also must state the number of patients in RISE-2 and a summary sentence of their combined clinical presentation (again also confirming that they have a LMN lesion), since from Ref.35 it seems that there are only 3 patients: the 1st one with a unilateral sciatic nerve lesion; the 2nd one with a bilateral D12/L1 cord medulla lesion giving mild spastic tremors; the 3rd one with a fractured left scapula and left brachial plexus lesion. Since the muscles studied in these three patients were, respectively :  Tibialis anterior (denervated Right, vs control Left), tibialis anterior (Bilaterally denervated), and Deltoid (denervated Left vs. control Right), the authors must here explain the limitations of extrapolation of the results of the RISE2-Italy Project to the more general situation of the thigh muscles in their main target patient group (bilateral complete conus and cauda equina syndrome, as in the 25 RISE patients). They should also clarify how the comparison of contraction-relaxation kinetics between the two sides in the two unilateral patients in RISE-2 could be made (line 245-247) if the stimulation required on the denervated side was at a high intensity level which would have been unacceptably painful on the control side.

Point 2. Rest of paper.
The clarification by the authors, (in their response to reviewer 2) that RISE and RISE2-Italy Projects comprise different patient groups, and the apparent low number of patients (3 patients) contributing to the RISE-2 study observations, makes it necessary to ask that for each of the different observations reported in this paper, the authors indicate how many patients were involved in each particular study, in order that the reader can gauge for themselves how much weight to allocate to each reported outcome (as below).     

Point 3.
Line 210-214.
Again here it would be helpful to know patient numbers in whom hamstring muscles were studied, particularly as the article in ref.23 is not accessible through PubMed or direct on an internet search.  Please could the authors again indicate this, and whether they are the same or  different patients from the original RISE cohort, perhaps as :  ‘cross sectional area of the Hamstrings improved in  the 25 RISE patients (or:  in n=x patients studied for this from the RISE cohort) , from a mean of 26.9+/-8.4 to 30.7+/-9.8 cm2 …’

Point 4.
Line 237-240.
  It would be helpful here to state the number of patients studied for skin changes, and whether they were from the 25 RISE patients :   eg. (from ref. 26).  ‘…On the other hand, studies on the skin in 13 patients from the RISE cohort, despite adding two additional years to duration from SCI, shows that hbFES recovers the properties of the epidermis to the same appearance as that presented in the skin biopsied earlier than one year after SCI [26].’

Point 5.
Line 307.
  In light of the comment above, it would be helpful here to say that this involved 16 patients.  eg.  ‘As described in Kern etal… 16 elderly patients performed….      

The rest of the points are mostly typographic ones :

Please see, and amend accordingly :

Line 34:  a muscle loss ofthat occurs

Line 39: The resulting t disuse atrophy

Line 41: and ofhigh medical expenses

Line 42: sarcopenia characterize complete     (characterizes)

Line 66-7:  ‘….is indeed responsible for a positive clinical result.’    Do the authors mean that the response of the non-targeted muscles is unexpectedly beneficial rather than being a problem, and if so, might this be better written as : ‘conversely, does in fact contribute to a positive clinical result’  

Line 94: has been confirmedby

Line 139: permanent denervatedmuscles

Line 153: the thigh, uby wetsponge-cloth fixed by elastic cuffs

Line 179: ‘there was a 1187% increase’.  ( just checking that there is no decimal point missing from here – ie. that a nearly 12x increase in force output is the correct order of magnitude )

Line 192: contractility in righboth legs,

Line 193-4: ‘consistently taken at the same position on both thighs, and at the same angle’   The authors have responded to the reviewer’s comment, but have not confirmed how the same consistent  positioning was achieved  ie. please indicate also the anatomical landmarks/measuring points used in relation to the thigh that ensured this consistency of positioning ?

Line 196: while 20% (4/20) acheaved the ability to stand     (achieved)

Line 234: Index (ID) an indexof the

Line 239-40:  ‘epidermis to those presented in the kin biopsied earlier’    See point 4 above : ‘epidermis to the same appearance as that presented in the skin biopsied earlier than one year after SCI [26].’

Line 253: Echomyography is a usefull tool     (useful)

Line 265: At 6 weeks post-SCI, muscle cross sectional areas weredecreased from 18% to 46%. (Should this be :  ‘were decreased by between 18% and 46 %’)

Line 267: weeksthe decreases

Line 274: undergoe   (undergo)

Line 295: is extended to Gluteal muscles to their training workout   (this is better written as ‘in their training workout’   or  ‘for their training workout’)

Line 276: dispalying

Line 296: this prolong the hours per day   (‘this prolongs’)

Line 313: muscle strenght

Author Response

We thanks the referee 2. for useful suggestions a implemented all suggestions (Point 1 to 5 and typographic suggestions).

Point 1.

As suggested we added:

The Rise2-Italy project was an independent extension of the EU RISE Project that enrolled three new patients with monolateral lower motor neuron denervation of skeletal muscles. In the three enrolled cases the contralateral innervated muscles were not submitted to electrical stimulation, but were utilizied as useful controls [35]. Monolateral denervated leg muscles were analyzed by Functional Echomyography to monitor thickness, contraction-relaxation properties and perfusion characteristics during and after electrical stimulation [35]. Morphology and ultrasonographic structure changed during several months of monolateral h-bFES from that of complete denervated muscle to the one that is “normal” in old patients. Contraction-relaxation kinetics, during electrical stimulation, showed a significantly longer relaxation phase in the monolateral denervated muscles.

Point 2. and POINT 3.

As suggested we indicated the number of enrolled patients in the EU RISE PROJECT adding:

4.2.1. Coactivation of the Hamstring muscles

Although not appreciated by bioengineers and physiatrists, who are more interested in selective electrical stimulation to control functional movements of arms and legs, the EU RISE Project demonstrated in the 25 enrolled patients that after two years of hbFES the QMC-CT cross sectional area of the Hamstrings improved, from 26.9+/-8.4 to 30.7+/-9.8 cm2, a significant 15% increase (p≤0.05), confirming that hbFES-induced muscle improvements [21], also occurred in Hamstrings (see details in Figure 4) [23].

Point 2. and POINT 4.

As suggested, we added:

4.2.2. SCI-induced skin atrophy recovers after two years of hbFES

The positive results of the hbFES approach on permanent lower motor neuron denervated leg muscles of the 25 patients enrolled in the EU Project RISE were extended to their skin by histology
and immunohistochemistry analyses in a sub-set of 13 patients of the EU Project RISE [26].

Using a grid the skin thickness was measured at regular intervals in cross sections of the biopsied skin. The mean value of the collected distances was taken as an estimate of the epidermal thickness.

Point 2. and POINT 5.

As suggested we added:

As described in Kern et al., 2014 [41], 16 elderly persons performed neuromuscular electrical stimulation training at home

REVISION OF ALL TYPOGRAPHICAL MISTAKES ARE ENLUGHTENED IN THE FILE OF REVISION R2.

Once again, we would like to thank Referee 2. for is patient and professional evaluation of our typescript and for the many useful suggestions.
